# Multi-Channel Masked Autoencoder and Comprehensive Evaluations for Reconstructing 12-Lead ECG from Arbitrary Single-Lead ECG

Jiarong Chen
Sun Yat-Sen University
Shenzhen, China
chenjr356@gmail.com

Wanqing Wu
Sun Yat-Sen University
Shenzhen, China
wuwanqing@mail.sysu.edu.cn

Shenda Hong
Peking University
Beijing, China
hongshenda@pku.edu.cn

## ABSTRACT

In the context of cardiovascular diseases (CVD) that exhibit an elevated prevalence and mortality, the electrocardiogram (ECG) is a popular and standard diagnostic tool for doctors, commonly utilizing a 12-lead configuration in clinical practice. However, the 10 electrodes placed on the surface would cause a lot of inconvenience and discomfort, while the rapidly advancing wearable devices adopt the reduced-lead or single-lead ECG to reduce discomfort as a solution in long-term monitoring. Since the single-lead ECG is a subset of 12-lead ECG, it provides insufficient cardiac health information and plays a substandard role in real-world healthcare applications. Hence, it is necessary to utilize signal generation technologies to reduce their clinical importance gap by reconstructing 12-lead ECG from the real single-lead ECG. Specifically, this study proposes a multi-channel masked autoencoder (MCMA) for this goal. In the experimental results, the visualized results between the generated and real signals can demonstrate the effectiveness of the proposed framework. At the same time, this study introduces a comprehensive evaluation benchmark named ECGGenEval, encompassing the signal-level, feature-level, and diagnostic-level evaluations, providing a holistic assessment of 12-lead ECG signals and generative model. Further, the quantitative experimental results are as follows, the mean square errors of 0.0178 and 0.0658, correlation coefficients of 0.7698 and 0.7237 in the signal-level evaluation, the average F1-score with two generated 12-lead ECG is 0.8319 and 0.7824 in the diagnostic-level evaluation, achieving the state-of-the-art performance. The open-source code is publicly available at https://github.com/CHENJIAR3/MCMA.

## CCS CONCEPTS

• **Computing methodologies** → *Artificial intelligence*; • **General and reference** → *General conference proceedings*; • **Applied computing** → **Health informatics**; • **Human-centered computing** → Mobile devices.

## KEYWORDS

12-lead ECG signals, Signal Reconstruction, Generated Models, Autoencoder, Evaluation Benchmark

**ACM Reference Format:**
Jiarong Chen, Wanqing Wu, and Shenda Hong. 2024. Multi-Channel Masked Autoencoder and Comprehensive Evaluations for Reconstructing 12-Lead ECG from Arbitrary Single-Lead ECG. In *Proceedings of Artificial Intelligence and Data Science for Healthcare (KDD-AIDSH).* ACM, New York, NY, USA, 11 pages. https://doi.org/10.1145/nnnnnnn.nnnnnnn

## 1 INTRODUCTION

Cardiovascular disease (CVD)[20, 21, 27] contributes to the leading mortality all around the world. In clinical practice, clinicians need to adopt some characterization tools[8] to diagnose cardiovascular disease, and one of the most popular tools is the standard 12-lead electrocardiogram (ECG). With the great development in deep learning, some researchers have trained a cardiologist-level model with the 12-lead ECG, like Ribeiro et al[25]. However, the 12-lead ECG signal collection process will put at least 10 electrodes on the surface, which would cause a lot of inconvenience and discomfort for users, and make long-term cardiac health monitoring difficult. As a consequence, researchers and markets are trying their best to explore some user-friendly devices for ECG signals collecting in the real-world application, including patch[12, 15, 33, 35], smartwatch[2, 10, 23, 28, 32], and armband[14, 18, 24]. Although the single-lead ECG can be used for cardiac abnormality classification, like the lead I for the Atrial Fibrillation[9], the lead V1 for the Brugada Syndrome[36], and the lead aVR for the Sinus Bradycardia[17], the full 12-lead ECG is necessary to provide comprehensive information, matching the knowledge and needs of clinical physicians, a limited or reduced number of leads ECG from wearable devices will not work effectively for doctors.

Consequently, to overcome these limitations, it is necessary to provide a direct approach to reduce the gap between the reduce-lead (Specifically, single-lead) and 12-lead ECG, that is, reconstructing 12-lead ECG with the reduced-lead ECG[1, 4–6, 11, 13, 19, 22, 29, 30], as seen in Fig.1. Although these methods can approximately reconstruct 12-lead ECG with the limited-lead ECG, there is a research gap to fill for this 12-lead ECG reconstruction task. Firstly, the traditional generative models usually focus on the fixed single-lead, and it is difficult to reconstruct 12-lead ECG with arbitrary single-lead ECG. Secondly, the related works[1, 5, 6, 11, 13, 29, 30] mainly focus on signal-level evaluation, instead of comprehensive evaluation for this task, and it is known the signal-level evaluation result will be

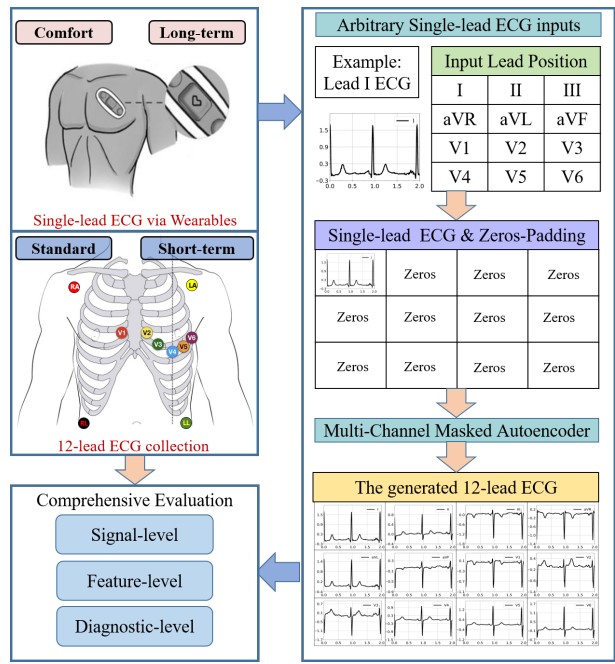

**Figure 1: The 12-lead ECG generation process with single-lead ECG, the input single-lead ECG could be arbitrary, including I, II, III, aVR, aVL, avF, V1, V2, V3, V4, V5, V6, and this case takes lead I as an example**

influenced by various noises. Therefore, the contributions in this study are as follows:

- A novel 12-lead ECG reconstruction framework, MCMA, is proposed in this study, and it can convert arbitary single-lead ECG into a 12-lead ECG.
- A benchmark for 12-lead ECG reconstruction tasks is proposed, ECGGenEval, including signal-level, feature-level, and diagnostic-level.
- The proposed framework can achieve state-of-the-art reconstruction performance in the internal and external test sets, with a mean square error of 0.0178 and a Pearson correlation coefficient of 0.7698.

This research article will be organized as follows. Section 1 presents the research motivation and contributions in this study. Section 2 will introduce the information about 12-lead ECG. Section 3 will introduce the proposed method, the public dataset, and the evaluation metric. Section 4 demonstrates the experimental results and discussions, which can prove the advantages over other studies. Section 5 will briefly conclude this work.

## 2 ECG BACKGROUND

The standard 12-lead ECG is one of the most popular schemes in clinical practice, which could provide enough cardiac health information. As mentioned, it is difficult to collect the long-term 12-lead ECG, since it causes a lot of inconvenience. This section will present the definition of 12-lead ECG, including time-domain and space-domain definitions. Firstly, a normal ECG should include P-wave,

QRS-complex, and T-wave, representing the corresponding cardiac activity. For example, the P-wave means the atrial depolarization, if the abnormal atrial depolarization process exists, there is a change in P-wave-based ECG. Secondly, the 12-lead ECG needs 10 electrodes on the surface, and each electrode position in the 12-lead ECG is seen in Table 1.

**Table 1: The electrode positions in 12-lead ECG**

| Lead | Electrode position |
|---|---|
| Lead I | Left Arm, Right Arm |
| Lead II | Left Foot, Right Arm |
| Lead III | Left Foot, Left Arm |
| Lead aVR | Right Arm |
| Lead aVL | Left Arm |
| Lead aVF | Left Foot |
| Lead V1 | The 4th intercostal space at the right sternal border |
| Lead V2 | The 4th intercostal space at the left sternal border |
| Lead V3 | The midpoint between V2 and V4 |
| Lead V4 | The 5th intercostal space at the midclavicular line |
| Lead V5 | Lateral to V4, at the left midaxillary line |
| Lead V6 | Lateral to V5, at the left midaxillary line |

According to Table 1, three limb leads belong to the bipolar lead, which requires two electrodes to collect the single-lead ECG. The nine remaining leads belong to the unipolar lead, and take the Wilson central terminal as the common reference.

## 3 METHOD

This section will introduce the details of the proposed framework, model architecture, ECG dataset, and evaluation metric. The model structure and setting is shown in Appendix.A.

### 3.1 MCMA Input & Output

The proposed framework will be named as Multi-Channel Masked Autoencoder (MCMA), which could mask different 11 leads and remain only a single-lead ECG to generate the entire 12-lead ECG. The input of MCMA is single-lead ECG, while the output of MCMA is 12-lead ECG. The signal length is 1024 for MCMA. In this study, no preprocessing step is used to avoid influencing ECG signals, like filtering or scaling.

### 3.2 Multi-Channel Configuration

In this study, MCMA, the proposed framework, needs to convert arbitrary single-lead ECG into the standard 12-lead ECG, and the multi-channel configuration will be used to reduce training and inference costs. On the one hand, with the multi-channel configuration, only one model is necessary to reconstruct 12-lead ECG from arbitrary single-lead ECG, distinguished from related works, like garg et al[5] focus on utilizing lead II to reconstruct 12-lead ECG, making it difficult for their model to work with other single-lead ECG as input. On the other hand, unlike Electrocardio panorama[3], the input ECG is one of the standard 12-lead ECG, and the output is the standard 12-lead ECG. Therefore, the input shape for MCMA is $(1024 \times 12)$, which could adapt different single-lead ECG as input.

## 3.3 Training MCMA

*3.3.1 Padding Strategy.* Since the proposed framework will be used to reconstruct 12-lead ECG with arbitrary single-lead ECG, and it is necessary to adopt a channel padding strategy in this study. To retain the space information for each single-lead ECG, the zero-padding strategy is proposed in this study. When the single-channel ECG will be processed into the 12-channel format, while the other channels will be set as zeros, as seen in Eq.1.

$$P(ecg_{12}, i) = I_z \times ecg_{12}[i] \tag{1}$$

In Eq.1, the shape of index matrix for zero-padding is $12 \times 1$, $I_z(i) = 1$ with other elements being zeros. Specifically, the output shape equals the input shape, and the shape of $ecg_{12}$ is $12 \times N$, then the shape of $ecg_{12}[i]$ is $1 \times N$, so the output shape also is $12 \times N$. With zero-padding, MCMA could adaptively solve different inputs. To highlight its advantages, the 12 copies for the single-lead ECG will be as a comparison, named as the copy-padding strategy. The index matrix for copy-padding strategy, $I_c$, all elements are 1. At the same time, the arbitrary input lead and the fixed lead (lead I) will be compared. In addition, the 12-lead ECG is provided in model training, and the padding strategy aims to mask the original 11-lead ECG with zeros or the remaining single-lead ECG in the standard 12-lead ECG. Meanwhile, only the single-lead ECG exists in the real-world application process, it should be with the padding strategy for the proposed framework.

*3.3.2 Loss Function.* The autoencoder could extract the latent representation with the raw data and convert the latent representation into the target output. The common loss function for autoencoder, $L$, can be shown in Eq.2.

$$L = ||ecg_{12} - AE(ecg_1)|| = ||ecg_{12} - AE(P(ecg_{12}, i))|| \tag{2}$$

In Eq.2, $AE$ denotes the autoencoder, and the 12-lead and single-lead ECG signals are represented by $ecg_{12}$ and $ecg_1$. $P$ means the padding strategy, including zero-padding and copy-padding. Additionally, $i$ means the index, varying from 1 to 12.

## 3.4 Inferencing MCMA

After the training process, MCMA would be used in real-world applications, i.e., the inferencing (testing) process. Firstly, the single-lead ECG should be provided as the input of MCMA. Secondly, the zeros-padding strategy will be used in this process, like Eq.1, but only the single-lead ECG inputs. Thirdly, the trained autoencoder could generate the reconstucted 12-lead ECG, which could be used to evaluation or downstream tasks. Then, the application process for MCMA could be seen in Eq.3.

$$g_{ecg} = AE(I_z \times ecg_1) \tag{3}$$

In Eq.3, $g_{ecg}$ is the generated 12-lead ECG with MCMA, $ecg_1$ is the single-lead ECG collected by wearable devices, $I_z$ could convert $ecg_1$ into the input of $AE$.

## 3.5 Comprehensive Evaluations

As mentioned, to fill the evaluation gap in this research field, this study will introduce a comprehensive evaluation benchmark, named as ECGGenEval. Specifically, this benchmark will contain 3 kinds of evaluation metrics, that is, signal-level, feature-level and diagnostic-level. The following contents will introduce the detailed evaluation metrics respectively.

*3.5.1 Signal-Level.* This study adopt the Pearson correlation coefficient ($PCC$) and mean square error ($MSE$) in the signal-level evaluation. It is necessary to define the real and generated ECG signal as $r_{ecg}$ and $g_{ecg}$. Then, the definitions for PCC and MSE are shown in Eq.4 and Eq.5.

$$PCC(r_{ecg}, g_{ecg}) = \frac{\mu(r_{ecg} \times g_{ecg}) - \mu(r_{ecg})\mu(g_{ecg})}{\sigma(r_{ecg})\sigma(g_{ecg})} \tag{4}$$

$$MSE(r_{ecg}, g_{ecg}) = \mu((r_{ecg} - g_{ecg})^2) \tag{5}$$

In these equations, as Eq.4 and Eq.5, $\mu(*)$ and $\sigma(*)$ denotes the mean value and standard deviation, respectively. The $PCC$ varies from -1 to 1, and the $MSE$ is at least bigger than 0. The relationship between $MSE$ and generation performance is positively related, while the relationship between $MSE$ and generation performance is negatively related. Therefore, based on the signal-level evaluation, a better generative model should be with a higher $PCC$ and lower $MSE$ from two different aspects.

*3.5.2 Feature-Level.* Although the signal-level evaluation is established, since the original ECG signals may be subject to varying degrees of influence from ambient noise, some metrics in other levels are needed for objectively evaluating the reconstruction performance. This section will introduce the heart rate for the feature-level evaluation. It is known that R-waves in real 12-lead ECG signals will theoretically occur simultaneously, and the generated signals should meet this requirement. Firstly, the mean heart rate ($MHR$) at the $j$th lead could be calculated, as shown in Eq.6.

$$MHR(j) = \frac{60 \times (n-1)}{\sum_{i=1}^{n-1}(R(i+1, j) - R(i, j))} \tag{6}$$

In Eq.6, the $i$th detected R-wave in $j$th lead will be denoted as $R(i, j)$, and it is expressed in seconds. Therefore, $MHR$ could represent the heartbeat per minute. Since the 12 heart rates are obtained, it is time to measure the heart rate consistency. In addition, some ECG signals collected in real-world applications will be difficult for R-wave detection of their enormous noise, so it is unnecessary to choose any lead as a reference. Based on the 12 $MHR$ from different 12-lead ECG, the average value $MMHR$ could be computed. Further, the feature-level evaluation in this 12-lead ECG reconstruction task will involve standard deviation ($SD$), Range (the difference between maximum and minimum), and coefficient of variation ($CV$), and they are expressed as $MHR_{SD}$, $MHR_{Range}$ and $MHR_{CV}$ respectively.

*3.5.3 Diagnostic-Level.* Further, this study also adopts the diagnostic-level evaluation for this 12-lead ECG reconstruction task. In the real-world application, there are some trained classifiers with 12-lead ECG as input, which are difficult to address some limited-lead ECG. The proposed framework could convert the limit-lead (even single-lead) ECG into 12-lead ECG, which bridges the limited-lead ECG to the trained classifiers. Therefore, it is necessary to evaluate the classification performance with the generated 12-lead ECG. The representative classification metric will be used in this study,

mainly including the precision (*Pre*), recall (*Rec*), specificity (*Spe*) and F1 score, as shown in literature[25].

## 3.6 Datasets

This study will adopt some public 12-lead ECG datasets to demonstrate the advantages for this study, including PTB-XL[31, 34], CPSC2018[16], and CODE-test[25]. The detailed descriptions of the datasets can be seen in Appendix.B.

## 4 RESULTS AND COMPARISON

The signal-level evaluation result is seen in Table 2, while the feature-level evaluation result in MCMA can be seen in Table 3, taking the internal testing dataset as an example. Based on the mentioned experimental results, it is known that the proposed framework could reconstruct high-fidelity 12-lead ECG with single-lead ECG. The average MSE and PCC in PTB-XL are 0.0178 and 0.7698, while the average MSE and PCC in CPSC2018 are 0.0658 and 0.7237, respectively. The experimental result of CPSC2018 will be seen in the following appendix, which could further demonstrate the effectiveness and advantage of the proposed MCMA.

Therefore, the reconstruction performance in the internal and external testing dataset could demonstrate its advantages, and MCMA could reconstruct the standard 12-lead ECG with arbitrary single-lead ECG as input. Therefore, the proposed method can provide a feasible solution when collecting the standard 12-lead ECG is inconvenient and difficult, like remote cardiac healthcare.Additionally, MCMA could convert arbitrary single-lead ECG into the standard 12-lead ECG. The comparisons in signal-level, feature-level, and diagnostic-level are shown in Table 4, Table 5 and Table 6, and the appendix will provide the comprehensive results.

Based on the results, it is known that the proposed framework can achieve state-of-the-art performance on the ECGGenEval, including the signal-level, feature-level, and diagnostic-level evaluation. For example, the MSE for generating 12-lead ECG with lead II is 0.0179, better than Grag et.al[5]. The internal and external testing set could prove its advance over other researchers. Therefore, MCMA can be used for 12-lead ECG reconstruction tasks while the single-lead ECG is collected, providing a novel solution in real-world cardiac healthcare applications. It is possible to improve the clinical importance of the wearable devices, playing an important role in ECG monitoring.

## 5 CONCLUSION

In conclusion, this study proposes a novel generative framework to reconstruct 12-lead ECG with single-lead ECG, as multi-channel masked autoencoder (MCMA), and it involves two main contributions. Firstly, unlike other methods, the proposed framework could convert arbitary single-lead ECG into the standard 12-lead ECG. The experimental results showed that the proposed framework had excellent performance, achieving state-of-the-art performance on the proposed benchmark, ECGGenEval, including the signal-level, feature-level, and diagnostic-level evaluation. For example, the average Pearson correlation coefficients in the internal and external testing set are 0.7698 and 0.7237, and it is shown that the zero-padding strategy could play an important role in the proposed framework. In the future, the proposed framework could be

adopted in clinical practice, which provides a novel feasible solution for long-term cardiac health monitoring.

## ACKNOWLEDGMENTS

This work was supported by National Natural Science Foundation of China (No. 62102008).

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

**Table 2: The signal-level evaluation of mean square error (MSE) and Pearson correlation coefficient (PCC) between the generated and real 12-lead ECG in the internal testing set, PTB-XL. Since MCMA could convert any single-lead ECG into 12-lead ECG, 12 different results are reported in this table.**

| Lead | I | II | III | aVR | aVL | aVF | V1 | V2 | V3 | V4 | V5 | V6 | Mean |
|---|---|---|---|---|---|---|---|---|---|---|---|---|---|
| | | | | | | | MSE | | | | | | |
| I | 0.0035 | 0.0099 | 0.0124 | 0.0035 | 0.0058 | 0.0104 | 0.0151 | 0.0465 | 0.0449 | 0.0306 | 0.0200 | 0.0142 | 0.0181 |
| II | 0.0076 | 0.0036 | 0.0113 | 0.0031 | 0.0087 | 0.0056 | 0.0167 | 0.0492 | 0.0480 | 0.0296 | 0.0183 | 0.0132 | 0.0179 |
| III | 0.0071 | 0.0079 | 0.0068 | 0.0059 | 0.0053 | 0.0053 | 0.0173 | 0.0505 | 0.0543 | 0.0393 | 0.0249 | 0.0171 | 0.0201 |
| aVR | 0.0055 | 0.0057 | 0.0166 | 0.0019 | 0.0091 | 0.0093 | 0.0151 | 0.0471 | 0.0443 | 0.0271 | 0.0157 | 0.0115 | 0.0174 |
| aVL | 0.0049 | 0.0098 | 0.0081 | 0.0053 | 0.0042 | 0.0076 | 0.0165 | 0.0485 | 0.0499 | 0.0368 | 0.0238 | 0.0164 | 0.0193 |
| aVF | 0.0080 | 0.0050 | 0.0082 | 0.0048 | 0.0071 | 0.0044 | 0.0174 | 0.0502 | 0.0523 | 0.0354 | 0.0225 | 0.0156 | 0.0192 |
| V1 | 0.0071 | 0.0099 | 0.0175 | 0.0047 | 0.0093 | 0.0114 | 0.0094 | 0.0366 | 0.045 | 0.0361 | 0.0228 | 0.0153 | 0.0187 |
| V2 | 0.0088 | 0.0108 | 0.0171 | 0.0059 | 0.0101 | 0.0114 | 0.0134 | 0.0215 | 0.0301 | 0.0351 | 0.0268 | 0.0179 | 0.0174 |
| V3 | 0.0082 | 0.0106 | 0.0179 | 0.0054 | 0.0098 | 0.0116 | 0.0156 | 0.0315 | 0.0171 | 0.0229 | 0.0225 | 0.0169 | 0.0158 |
| V4 | 0.0073 | 0.0089 | 0.0162 | 0.0044 | 0.0095 | 0.0103 | 0.0165 | 0.0427 | 0.0299 | 0.0128 | 0.0149 | 0.0137 | 0.0156 |
| V5 | 0.0067 | 0.0075 | 0.0161 | 0.0036 | 0.0093 | 0.0096 | 0.0164 | 0.047 | 0.0413 | 0.0198 | 0.0096 | 0.0104 | 0.0164 |
| V6 | 0.0065 | 0.0070 | 0.0161 | 0.0033 | 0.0093 | 0.0094 | 0.0159 | 0.0479 | 0.0459 | 0.0252 | 0.0125 | 0.0084 | 0.0173 |
| Mean | 0.0068 | 0.0081 | 0.0137 | 0.0043 | 0.0081 | 0.0089 | 0.0154 | 0.0433 | 0.0419 | 0.0292 | 0.0195 | 0.0142 | 0.0178 |
| | | | | | | | PCC | | | | | | |
| I | 0.9759 | 0.7604 | 0.5216 | 0.9116 | 0.813 | 0.5204 | 0.832 | 0.741 | 0.7358 | 0.8096 | 0.854 | 0.8676 | 0.7786 |
| II | 0.8349 | 0.9809 | 0.6089 | 0.9264 | 0.6254 | 0.8544 | 0.8045 | 0.7165 | 0.7111 | 0.8212 | 0.8761 | 0.8917 | 0.8043 |
| III | 0.8345 | 0.8017 | 0.9618 | 0.8097 | 0.8559 | 0.8715 | 0.7884 | 0.7012 | 0.6503 | 0.7276 | 0.7919 | 0.8111 | 0.8005 |
| aVR | 0.9048 | 0.8996 | 0.3094 | 0.9811 | 0.6043 | 0.6031 | 0.8346 | 0.7358 | 0.7444 | 0.8440 | 0.9035 | 0.9195 | 0.7737 |
| aVL | 0.9125 | 0.7406 | 0.8367 | 0.8327 | 0.9615 | 0.6851 | 0.8069 | 0.7233 | 0.6877 | 0.7505 | 0.8048 | 0.8219 | 0.797 |
| aVF | 0.8105 | 0.9148 | 0.8306 | 0.8516 | 0.7156 | 0.9672 | 0.7871 | 0.7045 | 0.6667 | 0.7599 | 0.8177 | 0.84 | 0.8055 |
| V1 | 0.8396 | 0.7501 | 0.2350 | 0.8570 | 0.5780 | 0.4618 | 0.9733 | 0.8154 | 0.7181 | 0.7524 | 0.8117 | 0.8396 | 0.7193 |
| V2 | 0.7905 | 0.7131 | 0.2801 | 0.8070 | 0.5498 | 0.4605 | 0.867 | 0.9732 | 0.8532 | 0.7654 | 0.7691 | 0.7911 | 0.7183 |
| V3 | 0.8100 | 0.7256 | 0.2358 | 0.8265 | 0.5509 | 0.4640 | 0.8231 | 0.875 | 0.9867 | 0.8831 | 0.8244 | 0.8157 | 0.7351 |
| V4 | 0.8405 | 0.7895 | 0.3260 | 0.8697 | 0.5692 | 0.5464 | 0.8066 | 0.7765 | 0.8657 | 0.9856 | 0.9164 | 0.8805 | 0.7644 |
| V5 | 0.8649 | 0.8346 | 0.327 | 0.9063 | 0.5857 | 0.5831 | 0.8107 | 0.7396 | 0.7713 | 0.9163 | 0.9833 | 0.9463 | 0.7724 |
| V6 | 0.8683 | 0.8515 | 0.3227 | 0.9174 | 0.5872 | 0.5981 | 0.8199 | 0.7273 | 0.7299 | 0.8667 | 0.9474 | 0.9804 | 0.7681 |
| Mean | 0.8572 | 0.8135 | 0.4830 | 0.8748 | 0.6664 | 0.6346 | 0.8295 | 0.7691 | 0.7601 | 0.8235 | 0.8584 | 0.8671 | 0.7698 |

**Table 3: The feature-level evaluation for the generated and real 12-lead ECG in the internal testing dataset PTBXL and the external testing dataset CPSC2018, including $MHR_{SD}$, $MHR_{CV}$, $MHR_{Range}$.**

| Input | PTBXL | | | CPSC2018 | | |
|---|---|---|---|---|---|---|
| | $MHR_{SD}$ | $MHR_{CV}$ | $MHR_{Range}$ | $MHR_{SD}$ | $MHR_{CV}$ | $MHR_{Range}$ |
| Original | 2.2137 | 3.21% | 7.2195 | 2.1313 | 2.65% | 7.1267 |
| I | 1.6276 | 2.41% | 5.0632 | 1.3633 | 1.81% | 4.3808 |
| II | 1.3492 | 2.00% | 4.0276 | 0.9823 | 1.33% | 3.0902 |
| III | 1.6850 | 2.43% | 5.1826 | 1.3029 | 1.73% | 4.1343 |
| aVR | 1.3884 | 2.09% | 4.1997 | 1.0209 | 1.39% | 3.2029 |
| aVL | 1.5817 | 2.31% | 4.8872 | 1.3417 | 1.77% | 4.312 |
| aVF | 1.4986 | 2.19% | 4.5079 | 1.1825 | 1.58% | 3.7303 |
| V1 | 1.3668 | 2.01% | 4.1093 | 1.1336 | 1.52% | 3.5885 |
| V2 | 1.4561 | 2.12% | 4.3951 | 1.1692 | 1.57% | 3.6568 |
| V3 | 1.4063 | 2.06% | 4.2434 | 1.1199 | 1.51% | 3.5077 |
| V4 | 1.3385 | 1.98% | 4.0322 | 1.0807 | 1.49% | 3.3710 |
| V5 | 1.3796 | 2.04% | 4.1673 | 1.0963 | 1.50% | 3.4172 |
| V6 | 1.3571 | 2.02% | 4.1031 | 1.0997 | 1.51% | 3.4346 |
| Mean | 1.4529 | 2.14% | 4.4099 | 1.1578 | 1.56% | 3.6522 |

**Table 4: The signal-level comparison in PTB-XL and CPSC2018, including Grag et al[5], Seo et al[29] and Joo et al[11]**

| Dataset | Metric | Method | Input | Value |
|---------|--------|--------|-------|-------|
| PTB-XL | MSE | Grag et al[5] | LeadII | 0.0292 |
| | | MCMA | Lead II | **0.0179** |
| | | Seo et al[29] | Lead I | 0.0279 |
| | | Joo et al[11] | Lead I | 0.0378 |
| | | MCMA | Lead I | **0.0181** |
| | PCC | Grag et al[5] | Lead II | 0.7981 |
| | | MCMA | Lead II | **0.8043** |
| | | Seo et al[29] | Lead I | **0.7885** |
| | | Joo et al[11] | Lead I | 0.7199 |
| | | MCMA | Lead I | 0.7786 |
| CPSC2018 | MSE | Grag et al[5] | Lead II | 0.0967 |
| | | MCMA | Lead II | **0.0662** |
| | | Seo et al[29] | Lead I | 0.0972 |
| | | Joo et al[11] | Lead I | 0.1118 |
| | | MCMA | Lead I | **0.0659** |
| | PCC | Grag et al[5] | Lead II | 0.7382 |
| | | MCMA | Lead II | **0.7616** |
| | | Seo et al[29] | Lead I | 0.7278 |
| | | Joo et al[11] | Lead I | 0.6845 |
| | | MCMA | Lead I | **0.7471** |

**Table 5: The feature-level comparison in PTB-XL and CPSC2018, including Grag et al[5], Seo et al[29] and Joo et al[11]**

| Dataset | Method | Input | $MHR_{SD}$ | $MHR_{CV}$ | $MHR_{Range}$ |
|---------|--------|-------|------------|------------|---------------|
| PTB-XL | Original | * | 2.2137 | 3.21% | 7.2195 |
| | Grag et al[5] | Lead II | **1.1608** | **1.70%** | **3.5872** |
| | MCMA | Lead II | 1.3492 | 2.00% | 4.0276 |
| | Seo et al[29] | Lead I | 1.8943 | 2.74% | 6.3984 |
| | Joo et al[11] | Lead I | 2.6891 | 4.03% | 8.8273 |
| | MCMA | Lead I | **1.6276** | **2.41%** | **5.0632** |
| CPSC2018 | Original | * | 2.1313 | 2.65% | 7.1267 |
| | Grag et al[5] | Lead II | **0.9545** | **1.24%** | **3.0523** |
| | MCMA | Lead II | 0.9823 | 1.33% | 3.0902 |
| | Seo et al[29] | Lead I | 2.1899 | 2.79% | 7.5269 |
| | Joo et al[11] | Lead I | 2.4136 | 3.31% | 8.1059 |
| | MCMA | Lead I | **1.3633** | **1.81%** | **4.3808** |

**Table 6: The diagnostic-level comparison in CODE-test, including Grag et al[5], Seo et al[29] and Joo et al[11]**

| Method | Input | $Pre$ | $Rec$ | $Spe$ | $F1$ |
|--------|-------|-------|-------|-------|------|
| * | 12-lead ECG | 0.8747 | 0.9100 | 0.9958 | 0.8872 |
| Grag et al[5] | Lead II | 0.7268 | **0.8542** | 0.9881 | 0.7808 |
| Input for MCMA | Lead II | 0.0682 | 0.0339 | 0.9778 | 0.0333 |
| MCMA | Lead II | **0.8099** | 0.7976 | **0.9935** | **0.7824** |
| Seo et al[29] | Lead I | 0.8248 | **0.8480** | 0.9948 | 0.8299 |
| Joo et al[11] | Lead I | 0.7817 | 0.7846 | 0.9938 | 0.7730 |
| Input for MCMA | Lead I | 0.3971 | 0.1309 | 0.9910 | 0.1824 |
| MCMA | Lead I | **0.8386** | 0.8381 | **0.9956** | **0.8319** |

Diagnosis and Mortality Risk Stratification. *European Heart Journal-Digital Health* (2024), ztae014.

[18] Jesús Lázaro, Natasa Reljin, Md-Billal Hossain, Yeonsik Noh, Pablo Laguna, and Ki H. Chon. 2020. Wearable Armband Device for Daily Life Electrocardiogram Monitoring. *IEEE Transactions on Biomedical Engineering* 67, 12 (2020), 3464–3473. https://doi.org/10.1109/TBME.2020.2987759

[19] Sidharth Maheshwari, Amit Acharyya, Pachamuthu Rajalakshmi, Paolo Emilio Puddu, and Michele Schiariti. 2014. Accurate and reliable 3-lead to 12-lead ECG reconstruction methodology for remote health monitoring applications. *IRBM* 35, 6 (2014), 341–350.

[20] Kevin Mc Namara, Hamzah Alzubaidi, and John Keith Jackson. 2019. Cardiovascular disease as a leading cause of death: how are pharmacists getting involved? *Integrated pharmacy research and practice* (2019), 1–11.

[21] Elizabeth G Nabel. 2003. Cardiovascular disease. *New England Journal of Medicine* 349, 1 (2003), 60–72.

[22] Stefan P Nelwan, Jan A Kors, Simon H Meij, Jan H van Bemmel, and Maarten L Simoons. 2004. Reconstruction of the 12-lead electrocardiogram from reduced lead sets. *Journal of electrocardiology* 37, 1 (2004), 11–18.

[23] Marco V Perez, Kenneth W Mahaffey, Haley Hedlin, John S Rumsfeld, Ariadna Garcia, Todd Ferris, Vidhya Balasubramanian, Andrea M Russo, Amol Rajmane, Lauren Cheung, et al. 2019. Large-scale assessment of a smartwatch to identify atrial fibrillation. *New England Journal of Medicine* 381, 20 (2019), 1909–1917.

[24] Vega Pradana Rachim and Wan-Young Chung. 2016. Wearable noncontact armband for mobile ECG monitoring system. *IEEE transactions on biomedical circuits and systems* 10, 6 (2016), 1112–1118.

[25] Antônio H Ribeiro, Manoel Horta Ribeiro, Gabriela MM Paixão, Derick M Oliveira, Paulo R Gomes, Jéssica A Canazart, Milton PS Ferreira, Carl R Andersson, Peter W Macfarlane, Wagner Meira Jr, et al. 2020. Automatic diagnosis of the 12-lead ECG using a deep neural network. *Nature communications* 11, 1 (2020), 1760.

[26] Olaf Ronneberger, Philipp Fischer, and Thomas Brox. 2015. U-net: Convolutional networks for biomedical image segmentation. In *Medical Image Computing and Computer-Assisted Intervention–MICCAI 2015: 18th International Conference, Munich, Germany, October 5-9, 2015, Proceedings, Part III 18*. Springer, 234–241.

[27] Gregory A Roth, Degu Abate, Kalkidan Hassen Abate, Solomon M Abay, Cristiana Abbafati, Nooshin Abbasi, Hedayat Abbastabar, Foad Abd-Allah, Jemal Abdela, Ahmed Abdelalim, et al. 2018. Global, regional, and national age-sex-specific mortality for 282 causes of death in 195 countries and territories, 1980–2017: a systematic analysis for the Global Burden of Disease Study 2017. *The Lancet* 392, 10159 (2018), 1736–1788.

[28] Alexander Samol, Kristina Bischof, Blerim Luani, Dan Pascut, Marcus Wiemer, and Sven Kaese. 2019. Recording of bipolar multichannel ECGs by a smartwatch: modern ECG diagnostic 100 years after Einthoven. *Sensors* 19, 13 (2019), 2894.

[29] Hyo-Chang Seo, Gi-Won Yoon, Segyeong Joo, and Gi-Byoung Nam. 2022. Multiple electrocardiogram generator with single-lead electrocardiogram. *Computer Methods and Programs in Biomedicine* 221 (2022), 106858. https://doi.org/10.1016/j.cmpb.2022.106858

[30] Jangjay Sohn, Seungman Yang, Joonnyong Lee, Yunseo Ku, and Hee Chan Kim. 2020. Reconstruction of 12-lead electrocardiogram from a three-lead patch-type device using a LSTM network. *Sensors* 20, 11 (2020), 3278.

[31] Nils Strodthoff, Patrick Wagner, Tobias Schaefter, and Wojciech Samek. 2021. Deep Learning for ECG Analysis: Benchmarks and Insights from PTB-XL. *IEEE Journal of Biomedical and Health Informatics* 25, 5 (2021), 1519–1528. https://doi.org/10.1109/JBHI.2020.3022989

[32] Geoffrey H Tison, José M Sanchez, Brandon Ballinger, Avesh Singh, Jeffrey E Olgin, Mark J Pletcher, Eric Vittinghoff, Emily S Lee, Shannon M Fan, Rachel A Gladstone, et al. 2018. Passive detection of atrial fibrillation using a commercially available smartwatch. *JAMA cardiology* 3, 5 (2018), 409–416.

[33] Mintu P Turakhia, Donald D Hoang, Peter Zimetbaum, Jared D Miller, Victor F Froelicher, Uday N Kumar, Xiangyan Xu, Felix Yang, and Paul A Heidenreich. 2013. Diagnostic utility of a novel leadless arrhythmia monitoring device. *The American journal of cardiology* 112, 4 (2013), 520–524.

[34] Patrick Wagner, Nils Strodthoff, Ralf-Dieter Bousseljot, Dieter Kreiseler, Fatima I Lunze, Wojciech Samek, and Tobias Schaeffter. 2020. PTB-XL, a large publicly available electrocardiography dataset. *Scientific data* 7, 1 (2020), 154.

[35] Yuki Yamamoto, Daisuke Yamamoto, Makoto Takada, Hiroyoshi Naito, Takayuki Arie, Seiji Akita, and Kuniharu Takei. 2017. Efficient skin temperature sensor and stable gel-less sticky ECG sensor for a wearable flexible healthcare patch. *Advanced healthcare materials* 6, 17 (2017), 1700495.

[36] Beatrice Zanchi, Francesca Dalia Faraci, Ali Gharaviri, Marco Bergonti, Tomas Monga, Angelo Auricchio, and Giulio Conte. 2023. Identification of Brugada syndrome based on P-wave features: an artificial intelligence-based approach. *Europace* 25, 11 (2023), euad334.

# A  PROPOSED MODEL

## A.1  Model Architecture

For this framework, it is necessary to design a proper model for 12-lead reconstruction, aiming to learn the effective representation. Firstly, it is necessary to take into account the locality and uncertainty of features within the ECG signals and there is no feature alignment in signal preprocessing. Therefore, the convolutional neural network can largely extract the local feature, which can play a key module in this 12-lead ECG reconstruction task. The designed model in MCMA is shown in Fig.2.

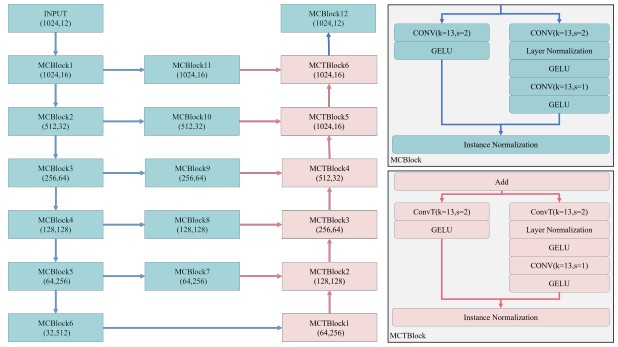

**Figure 2: The detailed model architecture, mainly including MCBlock and MCTBlock, k is for kernel size and s is for stride**

The designed model is motivated by ResNet[7] and UNet[26]. The whole model can be subdivided into two modules, namely, the downsampling and upsampling modules, which are composed of the multi-convolution block (MCBlock) and multi-convolution-transpose block (MCTBlock), respectively. The kernel size is 13 and the window size is 2. The activation function is GELU. To improve the gradient stability, layer normalization (LN) and instance normalization (IN) are used in each block. The skip connections could speed up the convergence rate of the model and improve the representation ability.

## A.2  Hyperparamaters

At the same time, the basic training recipe is provided in Table 7, including some hyperparamaters settings, like batch size and learning rate.

**Table 7: The hyperparameters configuration**

| hyperparameters | configuration |
|---|---|
| Batch size | 256 |
| Epochs | 100 |
| Signal Length | 1024 |
| Optimizer | Adam |
| Learning rate | 1e-3 |

# B  DATASET

## B.1  PTB-XL

In this study, PTB-XL[31, 34] will be used for model training, validating, and testing. As a large dataset, PTB-XL involves 21,799 clinical 10-second 12-lead ECG signals, and the sampling frequency is 500Hz. Based on the SCP-ECG standard, this dataset includes 71 kinds of ECG statements. As recommended, this study will adopt the cross-validation folds, in which the folds from the 1st fold to the 8th fold will be the training set, the 9th fold and the 10th fold act as the validation set and testing set, respectively. The ratio for training:validation: and testing is about 8:1:1.

## B.2  CPSC2018

To demonstrate the advantages and feasibility, CPSC2018[16] is used as an external testing set since the data distribution and information do not appear in model training and choosing. CPSC2018 contains 6,877 12-lead ECG, and these lengths varied from 6 seconds to 60 seconds with 500 Hz in sampling frequency. This dataset mainly includes a 9-type ECG, which aims to find out the cardiac arrhythmia detection tool.

## B.3  CODE-test

The above datasets just focus on similarity evaluation for the generated signals. This study will establish a benchmark for 12-lead ECG reconstruction. Therefore, this process requires representative research for 12-lead ECG classification, like Ribeiro et al[25]. This testing set is named CODE-test, including 827 12-lead ECG collected from different patients.

## B.4  Data Distribution

Since the generative model requires the 1024 point as input length, then the data distribution for two large-scale datasets could be listed as Table 8.

**Table 8: The data description of PTB-XL and CPSC2018**

| Dataset | Role | Number |
|---|---|---|
| PTB-XL | Training Set | 87200 |
|  | Validation Set | 10965 |
|  | Internal testing set | 11015 |
| CPSC2018 | External testing set | 55999 |

Table 8 presents the data distribution in PTB-XL and CPSC2018, including the internal and external testing set, which will be used for the signal-level and feature-level evaluation. Since CODE-test will be used in the diagnostic-level evaluation, it is necessary to introduce the ECG abnormality, as shown in Table 9

Based on Table9, there are six arrhythmia types in this dataset. It is important for the model to keep or generate pathological information. Therefore, the classification performance with the real 12-lead ECG will be adopted as the supremum, and the real single-lead ECG with padding strategy will be used as the baseline.

**Table 9: The data description of CODE-test**

| Abbreviation | Description | Quantity | Proportion |
| --- | --- | --- | --- |
| 1dAVb | 1st degree AV block | 28 | 3.4% |
| RBBB | right bundle branch block | 34 | 4.1% |
| LBBB | left bundle branch block | 30 | 3.6% |
| SB | sinus bradycardia | 16 | 1.9% |
| AF | atrial fibrillation | 13 | 1.6% |
| ST | sinus tachycardia | 36 | 4.4% |

## C  RESULTS

To evaluate the 12-lead ECG reconstruction performance for the proposed framework, a comprehensive evaluation benchmark is built in the mentioned content, named ECGGenEval. Therefore, the experimental results for MCMA are calculated in this section with the proposed benchmark, ECGGenEval, including the signal-level, feature-level, and diagnostic-level evaluation.

### C.1  Signal-Level

First of all, the signal-level evaluation is the primary evaluation metric, such as MSE and PCC. Unlike the traditional methods, this scheme has the advantage that arbitary single-lead ECG can be output to 12-lead ECG without training multiple generative models, effectively reducing the model training cost. The experimental results of MSE and PCC are shown in Table 2, where the horizontal direction represents the output and the vertical direction represents the input. Besides, the reconstruction performance in the external dataset, CPSC2018, is seen in Table 10.

### C.2  Feature-Level

To avoid the noise influence, and meet the clinical requirement, this study proposes the feature-level evaluation metric, including the standard deviation $MHR_{SD}$, Range $MHR_{Range}$ and coefficient of variation $MHR_{CV}$. The feature-level evaluation results in the internal testing set PTB-XL and external testing set CPSC2018 are shown in Table 3, respectively. Since the proposed method can convert arbitrary single-lead ECG into 12-lead ECG, these tables will involve 13 groups of experimental results, the first group will be the reference value as the original 12-lead ECG input.

Based on the experimental result, the conclusion can be drawn as following. For CPSC2018, the optimal result is from the generated 12-lead ECG by lead II ECG. The generated 12-lead ECG from arbitary single-lead ECG could produce a good heart rate consistency in different leads, and it can even be better than the original 12-lead ECG in some cases, due to the ECG signal denoising function in the proposed framework. Therefore, the feature-level evaluation can demonstrate the advantages of the MCMA. Additionally, the ECG morphological features in the single-lead would like to be calculated, demonstrating the similarity and consistency in the generated and real ECG signals. Taking lead I as example, Table 11 shows the difference between the generated from 12 different inputs and the original ECG, including the Heart rate(HR, bpm), P_amplitude(P_amp, mv), P_duration(P_dur, ms), PR_interval(PR, ms), QRS_amplitude(QRS_amp, mv), QRS_duration(QRS_dur, ms),

T_amplitude(T_amp, mv), ST(ms), QT_interval(QT, ms), QTc(ms), and Heart rate variation(ms). In this table, amp and dur represent the amplitude and duration, bpm means beat per minute, while ms and mv are millisecond and millivolt. As a reference, the calculated morphological feature values for the original lead I ECG are 74.2593bpm, 0.1159mv, 118.9081ms, 173.9268ms, 0.8535mv, 92.2338ms, 0.2123mv, 124.9686ms, 410.7288ms, 452.4418ms, 68.8162ms. Since the proposed framework could convert arbitary single-lead ECG into 12-lead ECG, Table 11 involves 12 groups of experimental result, including lead I ECG.

Based on Table 11, the ECG morphological features of the generated ECG are extremely close to that of the original real ECG. The duration similarity is better than amplitude similarity. Like the heart rate, the maximum difference is 0.8456 and 1.23% in lead aVF, which largely presents the feasibility of the proposed framework. Besides, it is necessary to study other leads with some cutting-edge technologies, and lead I is one of the mostly used cases in the standard 12-lead ECG. Besides the R-peak, it is difficult to find the morphological features with other lead ECG, especially the subtle feature. Therefore, ECGGenEval focuses on the mean heart rate in different leads ECG for feature-level evaluation.

### C.3  Diagnostic-Level

The downstream task can demonstrate the clinical importance of the generated 12-lead ECG, and the classifier is trained and validated by Ribeiro et al[25]. The proposed framework could generate 12-lead ECG with single-lead ECG, and the single-lead ECG could not be processed by a classifier trained with 12-lead ECG. For example, the classification performance of the generated 12-lead ECG with lead I is completed by the pretrained model[25], and the classifier could adopt the generated 12-lead ECG for arrhythmia classification. The average F1-score over 6 classes is 0.8319. Then, it is proven that MCMA could convert the single-lead ECG into the 12-lead ECG, and the generated 12-lead ECG can retain the pathological information, and it is different to the signal-level and feature-level evaluation. Therefore, with the multi-channel masked autoencoder, it is possible to complete arrhythmia classification with single-lead ECG. Additionally, the proposed framework can reconstruct 12-lead ECG with arbitary single-lead ECG. Further, the detailed diagnostic-level evaluations are shown in Table 12, including the original 12-lead ECG (as the reference), the single-lead ECG (i.e., MCMA input) and the generated 12-lead ECG (i.e., MCMA output), which shows the gain from the proposed framework.

According to Table 12, the classification performance of the generated 12-lead ECG is better than that of single-lead ECG and similar to the real 12-lead ECG, which can demonstrate the classification performance gain brought by MCMA. The generated 12-lead from lead I could provide the closest classification performance, the average $F1$ score is 0.8319, which exceeds other cases. From the view of the classification task, the classification performance in the above tables demonstrates the generated 12-lead ECG can be used for cardiac abnormality detection, which can prove its advantage in bridging the single-lead ECG and 12-lead ECG, and it is effective in generating the pathological information with single-lead ECG as input.

**Table 10: The signal-level evaluation of mean square error (MSE) and Pearson correlation coefficient (PCC) between the generated and real 12-lead ECG in the external testing set, CPSC2018**

| Lead | I | II | III | aVR | aVL | aVF | V1 | V2 | V3 | V4 | V5 | V6 | Mean |
|------|---|----|-----|-----|-----|-----|----|----|----|----|----|----|------|
| | | | | | | | MSE | | | | | | |
| I | 0.0256 | 0.0334 | 0.0345 | 0.0274 | 0.0269 | 0.0326 | 0.0567 | 0.0914 | 0.1022 | 0.1032 | 0.1179 | 0.1384 | 0.0659 |
| II | 0.0302 | 0.0268 | 0.0320 | 0.0263 | 0.0303 | 0.0272 | 0.0605 | 0.0971 | 0.1051 | 0.1034 | 0.1178 | 0.1379 | 0.0662 |
| III | 0.0307 | 0.0310 | 0.0275 | 0.0293 | 0.0267 | 0.0272 | 0.0609 | 0.0989 | 0.1133 | 0.1151 | 0.1274 | 0.1439 | 0.0693 |
| aVR | 0.0289 | 0.0299 | 0.0383 | 0.0241 | 0.0314 | 0.0313 | 0.0566 | 0.0937 | 0.1016 | 0.101 | 0.1155 | 0.1367 | 0.0657 |
| aVL | 0.0284 | 0.0345 | 0.0298 | 0.0296 | 0.0247 | 0.0305 | 0.0592 | 0.0943 | 0.1106 | 0.1154 | 0.1284 | 0.1449 | 0.0692 |
| aVF | 0.0315 | 0.0282 | 0.0287 | 0.0283 | 0.0286 | 0.0262 | 0.0626 | 0.0996 | 0.1111 | 0.1110 | 0.1249 | 0.1423 | 0.0686 |
| V1 | 0.0309 | 0.0346 | 0.0386 | 0.0281 | 0.0314 | 0.0337 | 0.0455 | 0.0802 | 0.1035 | 0.1133 | 0.1277 | 0.1438 | 0.0676 |
| V2 | 0.0315 | 0.0354 | 0.0387 | 0.0299 | 0.0313 | 0.0341 | 0.0544 | 0.0558 | 0.0811 | 0.1065 | 0.1295 | 0.1463 | 0.0645 |
| V3 | 0.0309 | 0.0342 | 0.0406 | 0.0291 | 0.0316 | 0.0335 | 0.0579 | 0.0684 | 0.0638 | 0.0907 | 0.1213 | 0.1426 | 0.0620 |
| V4 | 0.0303 | 0.0327 | 0.0406 | 0.0279 | 0.0319 | 0.0333 | 0.0596 | 0.0833 | 0.0787 | 0.0772 | 0.1120 | 0.1380 | 0.0621 |
| V5 | 0.0296 | 0.0323 | 0.0400 | 0.0269 | 0.0317 | 0.0332 | 0.059 | 0.0913 | 0.0943 | 0.0895 | 0.1000 | 0.1339 | 0.0635 |
| V6 | 0.0296 | 0.0322 | 0.0396 | 0.0266 | 0.0318 | 0.0331 | 0.0589 | 0.0955 | 0.1029 | 0.0977 | 0.1104 | 0.1247 | 0.0652 |
| Mean | 0.0298 | 0.0321 | 0.0357 | 0.0278 | 0.0299 | 0.0313 | 0.0577 | 0.0875 | 0.0974 | 0.1020 | 0.1194 | 0.1395 | 0.0658 |
| | | | | | | | PCC | | | | | | |
| I | 0.9671 | 0.7713 | 0.4671 | 0.8942 | 0.7086 | 0.5503 | 0.7277 | 0.6537 | 0.6914 | 0.8072 | 0.8586 | 0.8681 | 0.7471 |
| II | 0.7900 | 0.979 | 0.5957 | 0.9285 | 0.4784 | 0.8804 | 0.6703 | 0.6045 | 0.6693 | 0.8075 | 0.8614 | 0.8741 | 0.7616 |
| III | 0.7669 | 0.8115 | 0.9573 | 0.7981 | 0.8078 | 0.8752 | 0.6609 | 0.5923 | 0.5994 | 0.7108 | 0.7646 | 0.7821 | 0.7606 |
| aVR | 0.8715 | 0.9110 | 0.2685 | 0.9767 | 0.4172 | 0.6680 | 0.7228 | 0.6434 | 0.7084 | 0.8395 | 0.8985 | 0.9147 | 0.7367 |
| aVL | 0.8412 | 0.7250 | 0.8048 | 0.7829 | 0.9460 | 0.6755 | 0.6875 | 0.6341 | 0.6238 | 0.7113 | 0.7594 | 0.7756 | 0.7473 |
| aVF | 0.7409 | 0.9168 | 0.841 | 0.8418 | 0.6257 | 0.968 | 0.6339 | 0.5812 | 0.6184 | 0.7431 | 0.7870 | 0.8031 | 0.7584 |
| V1 | 0.7742 | 0.7329 | 0.2269 | 0.8151 | 0.3979 | 0.4976 | 0.9559 | 0.7495 | 0.6783 | 0.7281 | 0.7662 | 0.7907 | 0.6761 |
| V2 | 0.7301 | 0.6853 | 0.2112 | 0.7685 | 0.3921 | 0.4459 | 0.7724 | 0.9631 | 0.8494 | 0.7754 | 0.7501 | 0.7500 | 0.6745 |
| V3 | 0.7541 | 0.7376 | 0.1042 | 0.8111 | 0.3556 | 0.5020 | 0.7073 | 0.8577 | 0.9825 | 0.9000 | 0.8303 | 0.8107 | 0.6961 |
| V4 | 0.7838 | 0.7939 | 0.1143 | 0.8595 | 0.3475 | 0.5326 | 0.678 | 0.7320 | 0.8797 | 0.9833 | 0.9141 | 0.8781 | 0.7081 |
| V5 | 0.8218 | 0.8115 | 0.1602 | 0.8938 | 0.3788 | 0.5447 | 0.689 | 0.6655 | 0.7591 | 0.914 | 0.9787 | 0.9374 | 0.7129 |
| V6 | 0.8299 | 0.8147 | 0.1884 | 0.9052 | 0.3801 | 0.5535 | 0.691 | 0.6276 | 0.6944 | 0.8567 | 0.9353 | 0.9757 | 0.7044 |
| Mean | 0.8060 | 0.8075 | 0.4116 | 0.8563 | 0.5196 | 0.6411 | 0.7164 | 0.6921 | 0.7295 | 0.8147 | 0.8420 | 0.8467 | 0.7237 |

Therefore, based on the experimental results, it is concluded that MCMA could generate the standard 12-lead ECG with arbitrary single-lead ECG as input. At the same time, this study establishes a comprehensive evaluation benchmark, ECGGenEval, including the signal-level evaluation, feature-level evaluation, and diagnostic-level evaluation. The experimental results prove that MCMA can work well in ECGGenEval, which could be a baseline for future work. In the internal testing set, MCMA could perform a MSE of 0.0178 and a PCC of 0.7698, and the lowest MSE is 0.0156 in the lead V4 while the highest PCC is 0.8055 in the lead aVF. In the external testing set, MCMA could perform a MSE of 0.0658 and a PCC of 0.7237, and the lowest MSE is 0.062 in the lead V3, while the highest PCC is 0.7616 in the lead II. At the same time, for the feature-level evaluation, MCMA could get a $MHR_{SD}$ of 1.4529, a $MHR_{CV}$ of 2.10%, and a $MHR_{Range}$ of 4.4099 in the internal testing set, while the results are 1.1578, 1.56%, and 3.6522 in the external testing set. In the classification task, the highest $F_1$ is 0.8319, bridging the gap between the single-lead ECG and 12-lead ECG.

## C.4 Ablation Study

MCMA utilizes two key modules, one for arbitrary single-lead ECG reconstruction, and another for zero-padding strategy. Then, it is necessary to compare with different settings, including fixed-channel(lead I as an example) and copy-padding strategy. The signal-level evaluation metric, mean square error (MSE) and Pearson correlation coefficient (PCC) will be used. The experimental results comparison with different settings could be shown in Table 13, including the lead I and the average value for 12 single-lead ECG. In the most cases, MCMA has achieved an excellent result in the 12-lead ECG reconstruction task.

As Table 13 showing, the proposed framework is effective. Firstly, the multi-channel strategy could support arbitrary single-lead to generate 12-lead ECG. Although the reconstruction performance of lead I is slightly lower than the fixed-channel. When the lead I ECG inputs, the fixed-channel could have a MSE of 0.0176 and a PCC of 0.7885 better than MCMA, a MSE of 0.0181 and a PCC of 0.7786. However, for the fixed-channel, it is difficult to realize 12-lead ECG reconstruction with other leads, and the training and inference cost is largely different in training and storing 12 models with this

**Table 11: The feature-level evaluation of absolute and relative morphological feature difference between the generated and original lead I ECG**

| Input | HR | P_amp | P_dur | PR | QRS_amp | QRS_dur | T_amp | ST | QT | QTc | HRV |
|---|---|---|---|---|---|---|---|---|---|---|---|
| I | 0.5718 | 0.0369 | 7.1689 | 6.9264 | 0.2594 | 3.6948 | 0.0686 | 10.1397 | 10.6262 | 12.53 | 12.945 |
| % | 0.83% | 40.59% | 6.67% | 4.02% | 34.89% | 3.99% | 38.58% | 9.29% | 2.72% | 2.87% | 29.23% |
| II | 0.6144 | 0.0402 | 10.3392 | 11.0876 | 0.2876 | 8.495 | 0.0777 | 21.1166 | 18.8347 | 21.7852 | 15.5677 |
| % | 0.89% | 42.14% | 9.79% | 6.51% | 43.83% | 9.00% | 45.76% | 16.47% | 4.57% | 4.72% | 33.01% |
| III | 0.6612 | 0.0377 | 9.8265 | 10.2947 | 0.2861 | 7.8738 | 0.0744 | 18.8939 | 17.3506 | 19.8629 | 16.1653 |
| % | 0.98% | 37.87% | 9.23% | 6.02% | 43.01% | 8.32% | 42.44% | 15.48% | 4.25% | 4.34% | 35.83% |
| aVR | 0.7057 | 0.0393 | 9.5391 | 10.0386 | 0.2857 | 7.6208 | 0.074 | 17.9166 | 17.3683 | 20.2003 | 16.4932 |
| % | 1.05% | 39.81% | 9.07% | 5.93% | 42.57% | 8.11% | 41.73% | 14.48% | 4.24% | 4.41% | 42.87% |
| aVL | 0.7025 | 0.0412 | 10.0827 | 10.3801 | 0.2852 | 8.5999 | 0.0801 | 22.6017 | 19.9419 | 22.7184 | 16.0154 |
| % | 1.02% | 43.13% | 9.55% | 6.01% | 42.94% | 9.29% | 48.01% | 16.92% | 4.86% | 4.93% | 43.83% |
| aVF | 0.8456 | 0.0412 | 10.584 | 11.6172 | 0.2782 | 8.0677 | 0.087 | 24.6724 | 22.678 | 25.9355 | 20.782 |
| % | 1.23% | 42.18% | 10.00% | 6.58% | 41.67% | 8.61% | 54.04% | 19.48% | 5.52% | 5.62% | 84.92% |
| V1 | 0.6776 | 0.0406 | 8.4764 | 8.8283 | 0.2817 | 7.148 | 0.0719 | 15.8229 | 14.2657 | 16.4709 | 14.8274 |
| % | 0.99% | 43.68% | 7.89% | 5.15% | 41.23% | 7.66% | 42.11% | 12.52% | 3.49% | 3.61% | 31.62% |
| V2 | 0.8388 | 0.0377 | 9.4705 | 10.4393 | 0.2694 | 6.6889 | 0.0697 | 17.5462 | 16.8356 | 19.6939 | 18.2729 |
| % | 1.23% | 42.29% | 8.84% | 5.99% | 38.14% | 7.09% | 41.84% | 15.76% | 4.18% | 4.35% | 61.52% |
| V3 | 0.8084 | 0.0425 | 11.2461 | 11.3603 | 0.2804 | 8.8347 | 0.0921 | 25.7338 | 22.9555 | 26.3674 | 20.4351 |
| % | 1.19% | 42.60% | 10.94% | 6.56% | 42.56% | 9.59% | 57.04% | 19.09% | 5.60% | 5.75% | 63.80% |
| V4 | 0.6599 | 0.0404 | 10.7843 | 10.4909 | 0.294 | 8.0254 | 0.079 | 23.2426 | 21.0381 | 24.2838 | 16.5114 |
| % | 0.98% | 41.75% | 10.17% | 6.08% | 45.63% | 8.59% | 46.96% | 18.18% | 5.13% | 5.25% | 36.52% |
| V5 | 0.663 | 0.0416 | 11.7688 | 11.911 | 0.2909 | 9.1403 | 0.0827 | 24.2244 | 21.3619 | 24.411 | 16.0495 |
| % | 0.99% | 43.73% | 11.33% | 6.90% | 45.50% | 9.68% | 51.35% | 18.46% | 5.18% | 5.27% | 31.43% |
| V6 | 0.6571 | 0.0421 | 11.4455 | 11.8411 | 0.2885 | 8.8592 | 0.0827 | 23.5203 | 21.4196 | 24.5041 | 15.9346 |
| % | 0.95% | 45.04% | 11.04% | 6.94% | 44.48% | 9.43% | 50.78% | 18.10% | 5.24% | 5.33% | 30.94% |

setting. Secondly, the zero-padding strategy is better than the copy-padding strategy, while the two strategies both support the 12-lead reconstruction with arbitrary single-lead ECG. The mean MSE and PCC in MCMA are 0.0177 and 0.7697, while the mean MSE and PCC in copy-padding are 0.0195 and 0.7069. Therefore, based on the experimental result, MCMA is feasible to reconstruct 12-lead ECG with arbitrary single-lead ECG as input.

Received 2024-05-18; revised 2024-07-04; accepted 2024-06-29

**Table 12: The diagnostic-level evaluation of classification performance gain with the generated 12-lead ECG by MCMA**

| Input | Pre | Rec | Spe | F1 |
|---|---|---|---|---|
| Original 12-lead ECG[25] | 0.8747 | 0.9100 | 0.9958 | 0.8872 |
| I | 0.3971 | 0.1309 | 0.991 | 0.1824 |
| I+MCMA | 0.8386 | 0.8381 | 0.9956 | 0.8319 |
| MCMA Gain | 0.4415 | 0.7072 | 0.0046 | 0.6495 |
| II | 0.0682 | 0.0339 | 0.9778 | 0.0333 |
| II+MCMA | 0.8099 | 0.7976 | 0.9935 | 0.7824 |
| MCMA Gain | 0.7417 | 0.7637 | 0.0157 | 0.7491 |
| III | 0.1667 | 0.0056 | 0.9998 | 0.0108 |
| III+MCMA | 0.6983 | 0.6509 | 0.9915 | 0.6524 |
| MCMA Gain | 0.5316 | 0.6453 | -0.0083 | 0.6416 |
| aVR | 0 | 0 | 0.9985 | 0 |
| aVR+MCMA | 0.5005 | 0.4675 | 0.9770 | 0.4690 |
| MCMA Gain | 0.5005 | 0.4675 | -0.0215 | 0.4690 |
| aVL | 0 | 0 | 0.9998 | 0 |
| aVL+MCMA | 0.6265 | 0.637 | 0.9862 | 0.6136 |
| MCMA Gain | 0.6265 | 0.637 | -0.0136 | 0.6136 |
| aVF | 0 | 0 | 1 | 0 |
| aVF+MCMA | 0.5411 | 0.6057 | 0.9794 | 0.5167 |
| MCMA Gain | 0.5411 | 0.6057 | -0.0206 | 0.5167 |
| V1 | 0.2641 | 0.251 | 0.9973 | 0.2573 |
| V1+MCMA | 0.7579 | 0.8707 | 0.9921 | 0.8073 |
| MCMA Gain | 0.4938 | 0.6197 | -0.0052 | 0.5500 |
| V2 | 0.1667 | 0.0611 | 1 | 0.0894 |
| V2+MCMA | 0.7094 | 0.7983 | 0.9900 | 0.7410 |
| MCMA Gain | 0.5427 | 0.7372 | -0.01 | 0.6516 |
| V3 | 0.2428 | 0.1267 | 0.9909 | 0.1469 |
| V3+MCMA | 0.7654 | 0.8422 | 0.9927 | 0.7972 |
| MCMA Gain | 0.5226 | 0.7155 | -0.0063 | 0.6503 |
| V4 | 0.1667 | 0.009 | 1 | 0.0171 |
| V4+MCMA | 0.7900 | 0.8130 | 0.9929 | 0.786 |
| MCMA Gain | 0.6233 | 0.804 | -0.0071 | 0.7689 |
| V5 | 0 | 0 | 1 | 0 |
| V5+MCMA | 0.7918 | 0.8053 | 0.9931 | 0.7895 |
| MCMA Gain | 0.7918 | 0.8053 | -0.0069 | 0.7895 |
| V6 | 0.0833 | 0.0049 | 0.9996 | 0.0093 |
| V6+MCMA | 0.7514 | 0.7896 | 0.9921 | 0.7580 |
| MCMA Gain | 0.7918 | 0.8053 | -0.0069 | 0.7895 |

**Table 13: The module importance in MCMA**

| Channel | Padding Strategy | Input | Mean MSE | Mean PCC |
|---|---|---|---|---|
| Fixed- | Zeros | Lead I | 0.0176 | 0.7885 |
| Multi- | Copy | Lead I | 0.0191 | 0.7205 |
| Multi- | Zeros | Lead I | 0.0181 | 0.7786 |
| Fixed- | Zeros | 12 Single-lead | 0.0505 | 0.1342 |
| Multi- | Copy | 12 Single-lead | 0.0195 | 0.7069 |
| Multi- | Zeros | 12 Single-lead | 0.0177 | 0.7697 |