# OpenReview forum: "Multi-Channel Masked Autoencoder and Comprehensive Evaluations for Reconstructing 12-Lead ECG from Arbitrary Single-Lead ECG"
_KDD.org/2024/Workshop/AIDSH — KDD-AIDSH 2024 Poster_

### Official Review · Reviewer_bAQE · 2024-06-16
**Multi-Channel Masked Autoencoder and Comprehensive Evaluations for Reconstructing 12-Lead ECG from Arbitrary Single-Lead ECG**

**Rating:** 5
**Confidence:** 3

**Review:**

Review:

Using one lead to reconstruct the signals of other leads is indeed an interesting work, as it can effectively reduce the physical burden of wearable devices on the body. In this work, the authors used an Autoencoder as their waveform generator, training it by masking signals of other channels. From the results published by the authors, their model appears to be very effective, with an average MSE of about 0.01 on the PTB-XL dataset. This indicates that their model can almost capture the majority of ECG features. However, there are no other baselines for comparison in this task, so it's unclear whether other models could achieve the same effect. Additionally, the ECG length used in the experiments is 1024, meaning the model can only generate about 2 seconds of data on some high-frequency datasets, which limits its clinical use. However, I think increasing the length should be a simple task for this model. Overall, the method proposed in this article is good, but the experimental results and settings could be more detailed.

Strengths:

- The model is simple and easy to use.
- The experimental results look very good.

Weaknesses:

- There are no baseline methods for comparison for some of the generated results.
- The length of the generated signals limits its application in some clinical scenarios.
- Some experimental results are very strange.

Questions:
- I noticed that the model's input length is only 1024, while the sampling rate of common datasets is around 500Hz, meaning the model only inputs about 2 seconds of data. If the model could support longer reconstruction times, it could be more useful in downstream tasks.
- It is unclear whether the rows or columns are the input in Table 2.
- Can the authors provide some case studies to illustrate the effects of their proposed model?
- Did the authors preprocess the data but use normalized results when calculating outcomes? Since there are no baselines in the MSE experiments, it is unclear what insights the MSE experiment provides.
- Have the authors considered using other values for padding, such as 1, because using 0 as padding might primarily make the network's initial layers rely on bias, which seems counterproductive for loss reduction.
- Could the proposed method be effective on datasets with more arrhythmias? From a clinical standpoint, reconstructing heartbeat signals with abnormal patterns could help other algorithms diagnose more accurately.
- The Spe metirc in Table 12 appears negative.

---

### Official Review · Reviewer_LDBs · 2024-06-17
**Multi-Channel Masked Autoencoder and Comprehensive Evaluations for Reconstructing 12-Lead ECG from Arbitrary Single-Lead ECG**

**Rating:** 6
**Confidence:** 4

**Review:**

**Summary**
The paper designed multi-channel masked autoencoder (MCMA) to recover 12-lead ECG from arbitrary single-lead ECG. Further, to evaluate the generation task, the paper came up with a benchmark containing three evaluation perspectives, which are signal, feature and diagnostic levels. It shows that the proposed MCMA achieve sota performance on the proposed benchmark compared to baseline methods.

**Strengths**
- MCMA can rebuild 12-lead ECG from any single-lead ECG signal and achieve sota performance.
- Provide a comprehensive evaluation metrics for the ECG generation task.

**Weakness**
Weakness:
- Table 2 is a little bit confusing. I assume the table is presented to display the MSE and PCC of the model performance based on ECG with different lead choice. However, whether it is the row or the column that represents the input lead choice seems to be unspecified.
- Need to be consistent for the abbreviation of Pearson correlation coefficient. When it is first mentioned in the paper, it is abbreviated as CC. However, it is shortened as PCC in the tables.
- There are some typos. For instance, in the Introduction part, the sentence “However, the 12-lead ECG signal collection process will put at least 10 electrodes in the user’s surface” should be “on the surface” and “a limited or reduced number of leads ECG from wearable devices will don’t work effectively for doctors” should be “will not work”.

---

### Decision · Program_Chairs · 2024-06-28

Accept (Poster)